# Real-World Experiences Using Atezolizumab + Bevacizumab for the Treatment of Unresectable Hepatocellular Carcinoma: A Multicenter Study [note 1]

**DOI:** 10.3390/cancers17111814

**Published:** 2025-05-29

**Authors:** Maen Abdelrahim, Abdullah Esmail, Richard D. Kim, Sukeshi Patel Arora, Junaid Arshad, Ioannis A. Kournoutas, Conor D. O’Donnell, Todor I. Totev, Amie Tan, Fan Mu, Shravanthi M. Seshasayee, Sairy Hernandez, Nguyen H. Tran

**Affiliations:** 1Houston Methodist Research Institute, Houston, TX 77030, USA; aesmail@houstonmethodist.org; 2H. Lee Moffitt Cancer Center and Research Institute, Tampa, FL 33612, USA; richard.kim@moffitt.org; 3Mays Cancer Center, University of Texas Health San Antonio, San Antonio, TX 78229, USA; aroras@uthscsa.edu; 4University of Arizona Cancer Center, Tucson, AZ 85719, USA; junaidarshad@arizona.edu; 5Mayo Clinic, Rochester, MN 55905, USA; kournoutas.ioannis@mayo.edu (I.A.K.); tran.nguyen@mayo.edu (N.H.T.); 6Analysis Group, Inc., Boston, MA 02199, USA; todor.totev@analysisgroup.com (T.I.T.); fan.mu@analysisgroup.com (F.M.); shravanthi.seshasayee@analysisgroup.com (S.M.S.); 7Genentech, South San Francisco, CA 94080, USA; tan.ruoding@gene.com (A.T.); hernandez.sairy@gene.com (S.H.)

**Keywords:** atezolizumab, bevacizumab, clinical effectiveness, healthcare resource utilization, hepatocellular carcinoma, immunotherapy

## Abstract

Real-world evidence is important to help determine whether the effectiveness of therapies demonstrated in clinical trials is sustained in routine clinical practice, particularly in populations historically underrepresented in clinical trials. Although an emerging body of real-world evidence demonstrates the effectiveness of atezolizumab plus bevacizumab (A+B) as a first-line treatment for unresectable hepatocellular carcinoma, the characterization of treatment patterns, clinical outcomes, and safety-related healthcare resource utilization in United States (US)-only populations remains limited, especially among patients with impaired liver functions or those belonging to racial/ethnic minorities. This retrospective study assessed first-line A+B use among patients from five US institutions—the Mayo Clinic, Houston Methodist, Moffitt Cancer Center, Mays Cancer Center (University of Texas Health San Antonio), and the University of Arizona. The results confirm the real-world effectiveness of first-line A+B use in a diverse population in routine care. Patients with Child–Pugh class B reported similar toxicity-related discontinuation and hospitalization rates, warranting future prospective safety evaluation for A+B in these patients.

## 1. Introduction

The treatment paradigm for unresectable hepatocellular carcinoma (uHCC) has undergone considerable changes in the past two decades, evolving from tyrosine kinase inhibitors (TKIs; e.g., sorafenib and lenvatinib) to immunotherapy-based treatments as the current standard of care in the first-line (1L) setting [1]. Specifically, the United States (US) Food and Drug Administration (FDA)’s approval of atezolizumab plus bevacizumab (A+B) for the treatment of adult patients with uHCC who have not received prior systemic therapy in May 2020 marked a new era of immunotherapy-based treatment for 1L uHCC [1,2].

The approval of A+B for 1L uHCC was based on results from the global randomized phase 3 IMbrave150 (NCT03434379) trial [2,3,4], which showed that at the time of primary analysis, patients treated with A+B compared with sorafenib experienced significantly improved overall survival (OS; hazard ratio [HR] for death: 0.58; 95% confidence interval [CI]: 0.42–0.79) and progression-free survival (PFS; HR for progression or death: 0.59; 95% CI: 0.47–0.76) after a median follow-up of 8.6 months [3]. In addition, patients treated with A+B did not report new or unexpected adverse effects and showed a longer time to experience declining quality of life relative to patients treated with sorafenib [3]. On the basis of these results, the NCCN Clinical Practice Guidelines in Oncology (NCCN Guidelines^®^) were updated in 2021 to include A+B as a Category 1, Preferred 1L treatment option for eligible patients with uHCC and Child–Pugh (CP) class A (CP A) cirrhosis, and as a Category 2A option for those with CP B [5]; the NCCN Guidelines^®^ were subsequently updated, and 1L A+B is now recommended as a Category 1, Preferred 1L treatment option for eligible patients with uHCC regardless of CP status [6]. Of note, following the approval of A+B, the immunotherapy combination of tremelimumab plus durvalumab was also approved by the FDA in October 2022 for the treatment of 1L uHCC [7].

Using real-world data (RWD) is an important means to determine whether the effectiveness of therapies demonstrated in clinical trials is sustained in routine clinical practice as the latter typically follows less structured protocols and varies from trials with respect to outcome assessment timing and frequency, adherence monitoring, and complication management [8]. RWD are also more representative of the diverse patient population seen in routine care settings [8], including patients with uHCC typically excluded or underrepresented in clinical trials, such as those who are unfit (e.g., Eastern Cooperative Oncology Group Performance Status [ECOG PS] > 1), those with compromised liver function (e.g., CP B or CP C) [3,9], and those belonging to racial/ethnic minorities [10,11]. Thus, RWD could play a crucial role when weighing different treatment options for individual patients in clinical practice [12].

Although an emerging body of real-world evidence demonstrates the effectiveness of 1L A+B in uHCC [13,14,15,16,17,18,19,20,21,22], studies specific to US-only patient populations and including in-depth characterizations of their treatment patterns and clinical outcomes remain limited, particularly among patients with impaired liver functions or those belonging to racial/ethnic minorities. Furthermore, studies with insights into safety-related healthcare resource utilization (HCRU) in these populations are also scarce. To address these research gaps, the present study aimed to describe patient characteristics, treatment patterns, survival outcomes, and HCRU among a diverse cohort of patients with uHCC treated with 1L A+B from five academic medical centers across the US, including patients with impaired liver function and those belonging to racial/ethnic minorities.

## 2. Methods

### 2.1. Study Design and Data Source

This US-based, multicenter, non-interventional, retrospective chart review study involved the secondary use of de-identified data from medical charts of adult patients with uHCC treated with 1L A+B on or after 1 January 2019 within five medical centers in the US, including the Mayo Clinic Health System (all sites), Houston Methodist Research Institute (Houston, TX, USA), H. Lee Moffitt Cancer Center and Research Institute (Tampa, FL, USA), Mays Cancer Center, University of Texas Health San Antonio (San Antonio, TX, USA), and University of Arizona Cancer Center (Tucson, AZ, USA) [23]. Medical doctors and clinical research staff from participating medical centers retrospectively abstracted data from charts of all eligible patients between 8 May 2023 and 5 February 2024 using an electronic case report form.

The index date was defined as the date of initiation of A+B. The baseline period was defined as the 1-year period before the index date. The follow-up period spanned from the index date to the date of last contact or death, whichever occurred earlier.

Patient data were de-identified, and no protected health information was collected by the study sites. An Institutional Review Board (IRB) exemption was obtained from the independent Western-Copernicus Group IRB, which granted an exemption determination prior to the initiation of this study, per Title 45 of CFR, Part 46.104 (d) (4). Furthermore, where required, additional IRB exemptions were requested and approved at individual sites.

### 2.2. Study Population

The study population consisted of patients from the five aforementioned US medical centers who met the following criteria: (1) had a confirmed diagnosis of uHCC; (2) had no current or prior diagnosis of other primary malignancies (except prostate cancer, bladder cancer, and non-melanoma skin cancer) within 5 years of A+B initiation; (3) had initiated treatment with A+B for uHCC since 1 January 2019 outside of a clinical trial setting; and (4) were 18 years of age or older at the time of A+B initiation.

Patients were allowed to have initiated individual A+B regimens on the same date or on different dates, and the time between the initiation of atezolizumab and bevacizumab is summarized in the results.

### 2.3. Study Measures and Outcomes

Patient demographics, clinical characteristics (e.g., ECOG PS), and disease characteristics (e.g., primary hepatocellular carcinoma [HCC] etiology, CP class, Barcelona Clinic Liver Cancer [BCLC] stage, and albumin–bilirubin [ALBI] grade) were collected at or within 90 days prior to the index date. Comorbidities were assessed during the 1-year baseline period.

Time to treatment discontinuation (TTD) for A+B and reasons for treatment discontinuation were assessed. Patients were considered to have discontinued A+B treatment when both atezolizumab and bevacizumab were discontinued. Patients with ongoing treatment were censored at the date of last contact or date of death, as applicable.

The receipt of upper gastrointestinal esophagogastroduodenoscopy (EGD) during the 1-year baseline period and other HCC treatments received any time prior to and after A+B initiation were also assessed.

Real-world effectiveness outcomes, including clinician-assessed treatment response, real-world PFS (rwPFS), and OS, were assessed during the follow-up period. Clinician-assessed treatment response outcomes included (1) the proportion of patients with clinician-assessed best response, including real-world complete response (rwCR), real-world partial response (rwPR), real-world stable disease (rwSD), and real-world progressive disease (rwPD); (2) real-world overall response rate (rwORR), defined as the proportion of patients who had rwCR or rwPR of any duration out of all patients receiving 1L A+B; (3) real-world disease control rate (rwDCR), defined as the proportion of patients who had rwCR, rwPR, or rwSD out of all patients receiving 1L A+B; and (4) real-world duration of treatment response (rwDOR), defined as the duration of time from rwCR or rwPR until rwPD or the end of treatment, if no rwPD was observed. RwORR, rwDCR, and rwDOR were estimated based on documented clinician-assessed treatment responses obtained from up to eight radiological assessments in a patient’s chart following A+B initiation; the use of the Response Evaluation Criteria in Solid Tumors (RECIST) methodology was not required. RwPFS was defined as the time from the initiation of A+B to the first documented evidence of disease progression observed in routine clinical practice or death; patients who did not experience disease progression or death were censored at the date of last radiological assessment or the date of last contact. OS was defined as the time from the initiation of A+B to death from any cause. Death events were identified through structured electronic medical record (EMR) data, physician notes, discharge summaries, death certificates, obituary or Social Security Death Index searches, or secondary sources (e.g., confirmed by family) depending on the center; patients whose deaths were not captured were censored at the date of last contact.

HCRU (including the number and duration of hospitalizations, the number of emergency room [ER] visits, and reasons for HCRU) that were HCC-related based on clinical interpretation by the physicians was assessed over the first year following the initiation of A+B.

### 2.4. Subgroups

Select study outcomes were assessed in subgroups of interest. A “trial-like” subgroup of patients treated with 1L A+B were selected based on their similarity to patients enrolled in the IMbrave150 trial [4] with the following clinical characteristics: CP A, ECOG PS of 0–1, and ALBI grades 1–2. Exploratory subgroups included patients with CP A vs. CP B and patients stratified by race/ethnicity.

### 2.5. Statistical Analysis

Data from all centers were pooled for statistical analyses, cleaned, and reviewed for accuracy and clinical relevance. Study measures and outcomes were described by summary statistics using the number and percentage of patients in each category for discrete variables and the mean and standard deviation (SD) for continuous variables.

CP scores (5–15 points) and classes (A: 5–6 points; B: 7–9 points; and C: 10–15 points) were collected as available in patient charts. If the CP score was not available, a derived CP score was calculated during data analysis using the 1973 CP scoring rubric [24]. Similarly, BCLC staging data were collected from patient charts and further adjudicated with the corresponding patient’s CP class/score [25].

Clinician-assessed treatment response outcomes (rwORR, rwDCR, and rwDOR) were summarized as percentages. TTD, rwDOR, rwPFS, and OS were evaluated using Kaplan–Meier analysis, and medians and 95% CIs were reported. For TTD, patients with ongoing treatment were censored at the date of last contact or death, as applicable. For rwDOR, patients who did not experience disease progression or death were censored at date of last radiological assessment or date of last contact. For rwPFS, patients who did not experience disease progression or death were censored at the date of last radiological assessment or date of last contact. For OS, patients whose deaths were not captured were censored at the date of last contact.

All data were collected using electronic case report forms with required fields, minimizing missingness due to incomplete entries. Where applicable, “not available” or “unknown” response options were provided to explicitly capture unavailable data. Missing data were summarized as proportions and included in the denominator for descriptive analyses. Patients with missing endpoint data were excluded from relevant time-to-event analyses (one patient with a rwPR response did not have the assessment date of response documented and was not included in the analysis).

In subgroup analyses, statistical comparisons were performed using ANOVA tests for continuous variables, chi-squared tests for discrete variables, and log-rank tests for Kaplan–Meier analyses. For discrete variables with expected counts less than 10, Fisher’s exact tests were used instead of chi-squared tests. *p*-values < 0.05 were considered statistically significant.

Data management and analyses were conducted using Statistical Analysis Software (SAS^®^) for Windows release 9.4 (64-bit) or later (SAS Institute Inc., Cary, NC, USA) and RStudio (version 4.2 or later).

## 3. Results

### 3.1. Patient Characteristics

A total of 300 patients who received 1L A+B were included in the overall cohort. The mean age at treatment initiation was 67.4 (SD 9.3) years, and most patients were male (79.3%) and White (78.7%) (Table 1). Approximately 12% of patients had an ECOG PS of 2 or higher. During the baseline period, most patients were observed to have cirrhosis (69.0%) and hypertension (55.7%) (Table 2). Nearly half (48.0%) of the patients had hepatitis B or C during baseline, of whom 77.1% were treated with antivirals. The most frequent underlying HCC etiology was viral hepatitis (44.0%), primarily hepatitis C (Table 3). Liver function was classified as CP A in 73.0% of patients, CP B in 26.3% of patients, and CP C in 0.7% of patients. Most patients had BCLC stage C (82.3%) and ALBI grade 2 (57.7%), with 40.0% having experienced extrahepatic tumor spread and 25.3% having experienced Vp4 portal vein invasion (Table 3).

The characteristics of the subgroups are described in Appendix A. All patients in the “trial-like” subgroup (*n* = 194) had CP A (100.0%) and an ECOG PS of 0 (43.8%) or 1 (56.2%) per study design. In the subgroup of patients with CP A (*n* = 219) vs. CP B (*n* = 79), patients with CP B had more severely compromised liver function (CP A vs. B: cirrhosis: 60.7% vs. 91.1%; ascites: 7.3% vs. 45.6%; encephalopathy: 3.7% vs. 16.5%; esophageal varices: 13.7% vs. 31.6%), and there was a higher proportion of patients with bile duct invasion (1.8% vs. 11.4%) and an ECOG PS of 2 (8.2% vs. 20.3%) (all *p* < 0.01). In the subgroups of race/ethnicity, Black patients (*n* = 35) tended to have a higher proportion of CP A relative to White (*n* = 195), Hispanic or Latino (*n* = 39), and Asian (*n* = 21) patients, although this difference was not statistically significant.

### 3.2. Treatment Discontinuation

Over a median follow-up period of 8.7 (interquartile range: 4.0, 16.4) months, 93.3% of patients in the overall cohort were observed to have received bevacizumab within the first cycle of atezolizumab, with 85.7% receiving both drugs on the same day. By the end of the follow-up, 81.3% of patients discontinued A+B. The median TTD was estimated to be 5.3 months (95% CI: 4.6, 6.2; Figure 1). The reasons for discontinuation were primarily disease progression (atezolizumab: 47.5%; bevacizumab: 43.4%), followed by treatment toxicity/intolerance (atezolizumab: 13.9%; bevacizumab: 17.6%).

Patients with CP A had similar rates of discontinuation due to treatment toxicity/intolerance compared with those with CP B (CP A vs. B: atezolizumab: 12.0% vs. 17.3%; bevacizumab: 15.6% vs. 21.3%; both, *p* = 0.36). Relative to patients with CP A, a higher proportion of patients with CP B discontinued treatment due to death (CP A vs. B: atezolizumab: 1.8% vs. 8.0%; bevacizumab: 1.8% vs. 8.0%; both *p* < 0.05), consistent with the higher disease burden and severity expected in the CP B population.

### 3.3. Receipt of EGD and HCC Treatments Prior to and After A+B

At baseline, 74.7% of patients in the overall cohort had received ≥1 EGD procedures, among which 90.6% had received it in the 6 months prior to the initiation of A+B (Appendix A). Among patients with ≥1 EGD procedures at baseline, the procedures were performed mainly to assess eligibility for treatment with A+B (56.3%) or as a standard workup for HCC (33.9%). Esophageal varices were identified in 34.8% of patients who underwent an EGD procedure, and 34.7% of these patients were prescribed a corresponding treatment (Appendix A).

Prior to the initiation of A+B, over half of the patients in the overall cohort had received a locoregional HCC treatment (52.0%), including embolization (29.3%), and ablation (14.3%) (Table 4). Following treatment with A+B, most patients did not receive another systemic treatment (69.3%). Among those who did (*n* = 92), lenvatinib was the most common treatment received (60.9%; Table 4).

### 3.4. Real-World Effectiveness Outcomes

#### 3.4.1. Clinician-Assessed Treatment Response

During the follow-up period, 269 (89.7%) patients in the overall cohort had evaluable radiological response (Table 5). The most used radiological methods for assessing treatment response were computed tomography scans (59.0%) and magnetic resonance imaging (52.3%). As clinician-assessed best treatment response, rwCR was achieved in 7.3% of patients, rwPR in 23.3% of patients, and rwSD in 39.3% of patients. RwPD was observed in 19.7% of patients.

Based on the clinician-assessed best treatment responses, rwORR in the overall cohort was 30.7%, and rwDCR was 70.0%. Among patients achieving a rwCR or rwPR, the median rwDOR was 14.8 (95% CI: 10.3, not reached) months.

#### 3.4.2. RwPFS

In the overall cohort, 211 of 300 patients (70.3%) had evidence of disease progression or death during the follow-up, with a median rwPFS of 6.8 (95% CI: 5.8, 8.4) months (Figure 2). In the trial-like subgroup, 130 of 194 patients (67.0%) had evidence of disease progression or death during the follow-up, with a median rwPFS of 8.8 (95% CI: 7.6, 12.1) months.

Based on CP class, rwPFS was significantly different between patients with CP A and those with CP B (log-rank *p* < 0.001), with the median rwPFS being longer among patients with CP A (CP A vs. B: 8.3 [95% CI: 7.1, 11.1] vs. 3.1 [95% CI: 2.4, 5.8] months). Based on race/ethnicity, the difference in rwPFS was not statistically significant across groups (log-rank *p* = 0.26), although Black and Hispanic or Latino patients appeared to have longer median rwPFS (8.8 [95% CI: 4.1, not reached] months and 7.8 [95% CI: 4.7, 19.9] months, respectively) than other racial/ethnic groups (White: 6.8 [95% CI: 5.8, 8.4] months; Asian: 5.0 [95% CI: 2.1, 19.5] months).

#### 3.4.3. OS

In the overall cohort, 158 of 300 patients (52.7%) experienced death during the follow-up, with a median OS of 14.4 (95% CI: 12.3, 18.2) months (Figure 3). In the trial-like subgroup, 94 of 194 patients (48.5%) experienced death during the follow-up, with a median OS of 19.5 (95% CI: 14.6, 24.7) months. The 6-month OS rates in the overall cohort and in the trial-like subgroup were 74.5% and 86.4%, respectively; the 12-month OS rates were 56.1% and 66.4%, respectively; and the 24-month OS rates were 33.8% and 40.6%, respectively.

Based on CP class, OS was significantly different between patients with CP A and those with CP B (log-rank *p* < 0.001), with the median OS being longer among patients with CP A (CP A vs. B: 17.1 [95% CI: 14.4, 22.1] vs. 5.6 [95% CI: 4.6, 11.3] months). Based on race/ethnicity, OS was not statistically significantly different across groups (log-rank *p* = 0.29); the median OS tended to be longer among Black (28.0 [95% CI: 12.9, not reached] months) and Hispanic or Latino (20.5 [95% CI: 11.0, not reached] months) patients relative to White (14.1 [95% CI: 11.5, 17.1] months) and Asian (11.7 [95% CI: 7.4, not reached] months) patients.

### 3.5. HCC-Related Hospitalizations and ER Visits

During the first year of A+B initiation, 147 (49.0%) of patients in the overall cohort had ≥1 hospitalization (average length of hospitalizations: 5.8 [SD 4.7] days), 129 (43.0%) had ≥1 ER visit, and 287 (95.7%) had ≥1 oncology visit. HCC-related symptoms were the most common reason for hospitalizations/ER visits, with ≤10% of patients being hospitalized due to treatment-related adverse events (TrAEs; Appendix A).

Based on CP class, a significantly lower proportion of patients with CP A compared with CP B had ≥1 hospitalization (39.7% vs. 73.4%) or ER visit (37.9% vs. 57.0%) within the first year of A+B initiation (*p* < 0.001). Although not statistically significant, hospitalizations/ER visits due to TrAEs were comparable between patients with CP A and CP B, whereas the frequency of hospitalizations/ER visits due to HCC-related symptoms or disease progression appeared to be numerically higher among patients with CP B, in line with the higher disease burden and severity expected in the CP B population. HCRU was not statistically different by race/ethnicity.

## 4. Discussion

This US multicenter chart review study assessed the effectiveness of 1L A+B among patients with uHCC treated in routine clinical settings, which could differ from clinical trials with respect to outcome assessment, adherence monitoring, and complication management [8]. The patient population in this study was broad and diverse in terms of age, ECOG PS, liver function, and race/ethnicity. Many patients displayed clinical characteristics that would have excluded them from participating in the IMbrave150 trial [26], such as ECOG PS ≥ 2 and impaired liver function (e.g., more than one in four patients had CP class B or C cirrhosis). Among this diverse real-world patient population, the median rwPFS was 6.8 months, and the median OS was 14.4 months. Notably, among the “trial-like” subgroup of patients in this study who had clinical characteristics similar to those in the IMbrave150 trial, real-world clinical effectiveness appears similar to that reported from the trial (median rwPFS of 8.8 months vs. median PFS of 6.8 months in trial; median OS of 19.5 months vs. 19.2 months in trial). Our study also suggests that treatment with A+B may be well tolerated, as evidenced by the low proportion of patients (14–18%) who discontinued A+B due to treatment toxicity/intolerance, and that less than 10% of hospitalizations and ER visits during the first year after A+B initiation were TrAE-related. Collectively, the findings from this study provide further evidence to support the effectiveness of 1L A+B for uHCC in appropriately selected patients in clinical practice.

The clinical and disease characteristics of patients in this study are largely aligned with real-world uHCC populations receiving A+B reported in the existing literature [27]. In the present study, more patients had an underlying viral (44%) than non-viral HCC etiology (36%), consistent with the findings in the 2023 meta-analysis conducted by Kulkarni et al. including 46 real-world studies on 1L A+B in uHCC, which reported 60% viral and 40% non-viral etiology [27]. Of note, while 87% of HCC cases with an underlying viral etiology in our study were caused by hepatitis C, the more predominant cause observed in both the meta-analysis and the IMbrave150 trial was hepatitis B. This difference in viral etiology may reflect the difference in etiology seen between US and ex-US populations; notably, a recent post hoc analysis of IMbrave150 observed no differences in ORR, PFS, or OS across subgroups with varying HCC etiologies [28]. Meanwhile, patients involved in Kulkarni et al.’s meta-analysis and those in our study had, on average, similar cancer stage (BCLC stage C: 66% vs. 82%) and level of daily functioning (ECOG PS 0 or 1: 91% vs. 86%) [27], suggesting that our study population largely reflects the characteristics of patients with uHCC receiving A+B in the real world.

The 1L A+B discontinuation rate (81%) and median TTD (5.3 months) in the present study are comparable to those observed in other real-world analyses of patients with uHCC [15,16]. For instance, in a 2022 US-based retrospective claims study, 72% of patients with uHCC discontinued 1L A+B after a median follow-up of 15.3 months, with an estimated median TTD of 5.1 months [15]. In contrast, the median TTD in the present study appears shorter than that in the IMbrave150 trial, which reported a median TTD of 7.4 months [4], potentially owing to differences in clinical assessment criteria and subsequent treatment protocols in clinical trials compared to routine medical practice. Following 1L A+B discontinuation, roughly 30% of patients in the present study initiated a different systemic therapy. While this proportion appears to vary in prior real-world studies (19–43%), the most observed subsequent systemic therapy across prior studies was TKIs, primarily lenvatinib [15,16], which is consistent with the observations in this study.

Effectiveness outcomes in the present study were obtained based on treatment responses assessed by clinicians as opposed to those obtained using RECIST or modified RECIST criteria as in the IMbrave150 trial and some real-world studies [3,4,27]. Nevertheless, the treatment response outcomes in this study (rwORR: 31%; rwDCR: 70%; rwDOR: 14.8 months) are comparable to those in the trial (ORR: 30%; DCR: 74%; DOR: 18.1 months) [4] and in Kulkarni et al.’s meta-analysis (ORR: 29%; DCR: 77%) [27] assessed using RECIST criteria. The survival outcomes in the present study also align with those in the IMbrave150 trial, especially among the “trial-like” subgroup, as well as other real-world studies among patients with uHCC receiving 1L A+B [4,18,19,20,27]. In a prospectively maintained global uHCC registry study, the median PFS was 6.9 months, and the median OS was 15.7 months [19]. In a retrospective study based on EMR data from the US Oncology Network, the median PFS and OS were 6.4 and 13.2 months, respectively [18]. In another EMR-based US study among veterans, the median OS was 12.8 months [20].

OS in real-world studies is generally shorter than that in the IMbrave150 trial [4,18,19,20,27], which may be partly attributable to the expanded use of A+B in patients outside of the trial’s eligibility criteria, particularly pertaining to the degree of liver dysfunction. Given the higher proportion of patients with CP B in routine medical practice relative to the trial and the lack of alternative treatments, real-world studies have shown worse survival outcomes among patients with CP B than those with better liver function [18,20,21,27]. The present study, which included 26% of patients with CP B, also found that both rwPFS and OS were longer for patients with CP A than those with CP B (8.3 vs. 3.1 months and 17.1 vs. 5.6 months, respectively), in line with the findings in Kulkarni et al.’s meta-analysis and other real-world US studies [18,20,21,27]. For instance, in the US Oncology Network study, rwPFS and OS among patients with CP A vs. CP B were 7.3 vs. 5.7 months and 16.5 vs. 7.5 months, respectively [18]. In a multicenter EMR-based US study assessing the survival outcomes associated with 1L A+B by baseline liver function, patients with CP A compared with those with CP B exhibited significantly longer PFS (8.9 vs. 4.8 months) and OS (21.6 vs. 6.4 months) [21]. Together with the literature, our findings appear in line with the known underlying complications and poor prognosis associated with CP B disease [27], which may explain the shorter OS in the present overall cohort compared to the IMbrave150 trial.

Few studies have examined effectiveness outcomes associated with 1L A+B in uHCC among racial/ethnic minorities. Although the findings of the present study should be considered exploratory given the small sample sizes in the racial/ethnic subgroups, our analyses suggested numerically longer survival outcomes in Black patients and Hispanic patients compared with White patients. These findings align with a similar real-world study of US Veterans with uHCC [20] in which a numerically larger OS benefit of 1L A+B over select TKIs was observed in Black and Hispanic patients compared with White patients, although this trend was not statistically significant in the adjusted analysis. Future studies with larger sample sizes are warranted to evaluate the effectiveness of 1L A+B in various racial/ethnic groups.

To our knowledge, this study is among the first to assess HCRU in patients receiving 1L A+B in routine clinical practice, including the reasons for hospitalizations or ER visits. Our findings demonstrate that about half of the patients receiving 1L A+B were hospitalized or had an ER visit within the first year of treatment, primarily owing to HCC-related symptoms, with less than 10% due to TrAEs. Although there is a paucity of relevant prior studies in the literature on HCRU for comparisons, our findings are in line with the results from the above-mentioned prospectively maintained global uHCC registry study in which patients with CP A and those with CP B reported similar rates of TrAEs attributable to either drug, suggesting similar tolerability to A+B treatment in both patient groups regardless of liver function [16]. Our HCRU findings corroborate this observation, as comparable rates of hospitalizations and ER visits due to TrAEs between the CP A and CP B subgroups were observed. The ongoing prospective KIRROS trial (NCT06096779 [29]) will help further characterize the safety of 1L A+B in CP B patients.

The findings of the present study should be interpretated in light of several limitations. As a study that used RWD that were collected as part of routine clinical practice and not for research purposes, data collection efforts were bound by the availability of data in a patient’s chart, and the measurement of outcomes would not have been standardized. For example, it was not feasible to determine whether specific TrAEs required steroid treatment or were attributable to bevacizumab versus atezolizumab. Furthermore, any health-related data collected outside of the clinical sites (e.g., in a primary care or emergency setting) may not be captured in the data. A pertinent challenge in our study was the lack of uniform availability of CP score/class and missing values for the labs/disease severity components of the CP algorithm. To the extent possible, we relied on the clinical judgment of the research staff and investigators at the participating sites to provide estimated inputs to resolve missing data, minimizing the impact of the missing values. Meanwhile, clinician-assessed treatment responses may vary from responses assessed using the standardized RECIST criteria, resulting in potential differences in response rates compared to clinical trials. To avoid confusion, relevant outcomes assessed in this study (e.g., ORR, DCR, DOR, and PFS) were reported with the prefix “real-world (rw)” to distinguish them from corresponding clinical trial endpoints derived based on the RECIST criteria. Lastly, patient data included in this study were based on routine clinical practice, and thus, disease and treatment characteristics were assessed at the physicians’ discretion with varying frequency and may also differ across clinical sites. Nonetheless, training was provided for all assigned research staff and investigators at the initiation of data collection to ensure the electronic case report forms were completed as accurately as possible.

## 5. Conclusions

This study suggests that A+B could be an effective treatment in routine clinical practice. Many patients in this real-world US cohort had clinical features such as ECOG PS ≥ 2 and CP class B or C cirrhosis that would have excluded them from the IMbrave150 trial. In addition, patients treated in real-world clinical practice are more racially diverse and receive care that varies from clinical trial protocols in terms of monitoring frequency and timing, treatment adherence, and complication management. Despite these differences, many of which could have led to less favorable outcomes, the patients demonstrated a median rwPFS of 6.8 months and OS of 14.4 months. Although patients with CP B had higher hospitalization rates due to disease progression or symptoms, consistent with their underlying liver complications, they had similar rates of toxicity-related treatment discontinuation and TrAE-related hospitalizations compared to patients with CP A. Further characterization of safety for A+B in CP B patients within a prospective controlled trial is warranted and underway.

## Figures and Tables

**Figure 1 cancers-17-01814-f001:**
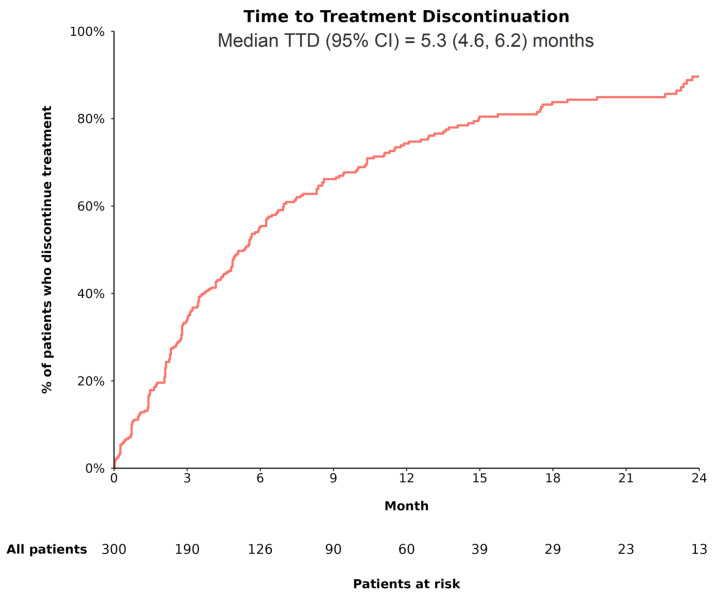
TTD among patients with uHCC receiving A+B as 1L treatment. **Abbreviations**—1L: first line; A+B: atezolizumab plus bevacizumab; CI: confidence interval; TTD: time to treatment discontinuation. **Notes:** 1. Patients were considered to have discontinued treatment when both atezolizumab and bevacizumab were discontinued. 2. Time to treatment discontinuation (i.e., time from initiation of 1L A+B to its discontinuation) was estimated using Kaplan–Meier analysis as one minus corresponding survival function. Patients with ongoing treatment were censored at date of last contact or date of death, as applicable.

**Figure 2 cancers-17-01814-f002:**
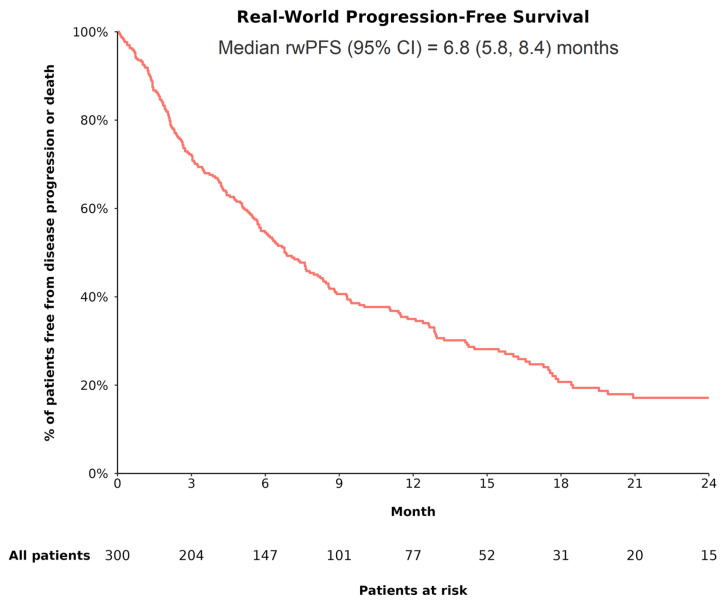
rwPFS among patients with uHCC receiving A+B as 1L treatment. **Abbreviations**—1L: first line; A+B: atezolizumab plus bevacizumab; CI: confidence interval; rwPFS: real-world progression-free survival. **Note:** RwPFS (the time from the initiation of treatment with A+B to the first documented evidence of disease progression observed in routine clinical practice or death) was estimated using Kaplan–Meier analysis. Patients who did not experience disease progression or death were censored at the date of the last radiological assessment or date of last contact.

**Figure 3 cancers-17-01814-f003:**
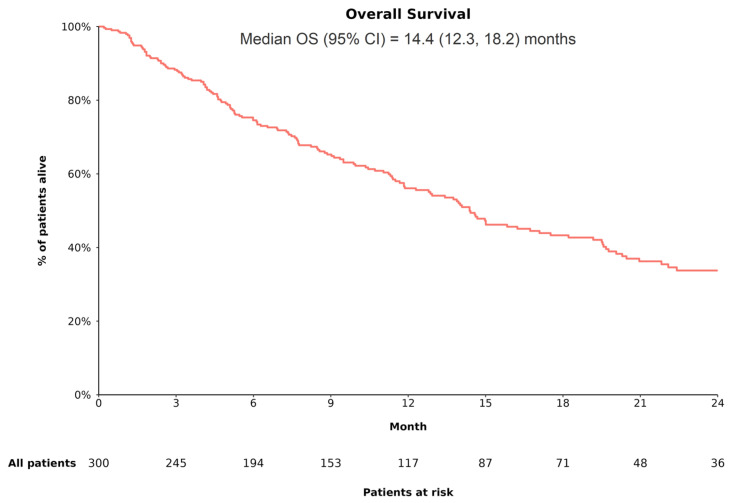
OS among patients with uHCC receiving A+B as 1L treatment. **Abbreviations**—1L: first line; A+B: atezolizumab plus bevacizumab; CI: confidence interval; OS: overall survival. **Note:** OS (the time from the initiation of treatment with A+B to death from any cause) was estimated using Kaplan–Meier analysis. Patients for whom death was not captured were censored at the date of last contact.

**Table 1 cancers-17-01814-t001:** Demographic characteristics of patients with uHCC receiving A+B as 1L treatment.

	Overall Cohort
	*n* = 300
**Medical center where patients were treated, *n* (%)**	
Mayo Clinic	142 (47.3%)
Houston Methodist	76 (25.3%)
Moffitt Cancer Center	41 (13.7%)
Mays Cancer Center	29 (9.7%)
University of Arizona	12 (4.0%)
**Patient age at initiation of 1L A+B ^1^ (mean ± SD; years)**	67.4 ± 9.3
**Male, *n* (%)**	238 (79.3%)
**Race ^2,3^, *n* (%)**	
White	236 (78.7%)
Black or African American	35 (11.7%)
Asian	18 (6.0%)
Other ^4^	5 (1.7%)
**Ethnicity ^3^, *n* (%)**	
Hispanic or Latino	39 (13.0%)
Not Hispanic or Latino	255 (85.0%)
**Education, *n* (%)**	
High school diploma or less	49 (16.3%)
Some college or associate’s degree	42 (14.0%)
College graduate/bachelor’s degree or advanced degree	55 (18.3%)
Unknown	154 (51.3%)
**Employment status ^2^, *n* (%)**	
Employed	51 (17.0%)
Unemployed ^5^	22 (7.3%)
Retired	157 (52.3%)
On disability ^6^	37 (12.3%)
Unknown	47 (15.7%)
**Insurance type ^2,3^, *n* (%)**	
Medicare	204 (68.0%)
Commercial/private insurance	95 (31.7%)
Medicaid	21 (7.0%)
Other insurance ^7^	12 (4.0%)

**Abbreviations**—1L: first line; A+B: atezolizumab plus bevacizumab; SD: standard deviation; uHCC: unresectable hepatocellular carcinoma. **Notes:** ^1^ For patient privacy, exact ages of patients aged ≥90 years were not collected. Age at initiation of 1L A+B treatment for these patients was set to 90 years by default in calculation of summary statistics. ^2^ Multiple responses were allowed for this question, and percentages may add up to more than 100%. ^3^ Percentages may not add up to 100% because small proportion of patients with unknown information has not been reported in table. ^4^ Other race categories were American Indian or Alaska Native or Hawaiian. ^5^ “Unemployed” category included patients who were unemployed or homemakers. ^6^ “On disability” categories were allowed to be selected together with all other categories except “Unknown”. ^7^ Other insurance categories included military insurance (Veterans Affairs or active military).

**Table 2 cancers-17-01814-t002:** Clinical characteristics of patients with uHCC receiving A+B as 1L treatment.

	Overall Cohort
	*n* = 300
**ECOG PS ^1^ within 90 days of initiation of 1L A+B treatment, *n* (%)**	
Grade 0: Fully active	110 (36.7%)
Grade 1: Restricted	147 (49.0%)
Grade 2: Ambulatory	34 (11.3%)
Grade 3: Confined to bed or chair	1 (0.3%)
**Comorbidities, assessed during baseline ^2^, *n* (%)**	
Cirrhosis	207 (69.0%)
Hypertension	167 (55.7%)
Diabetes mellitus	113 (37.7%)
Esophageal varices	55 (18.3%)
Ascites	54 (18.0%)
Mild to moderate ascites	48 (88.9%)
Severe ascites	6 (11.1%)
Encephalopathy	22 (7.3%)
Encephalopathy grade 1 or 2	13 (59.1%)
Encephalopathy grade unknown	9 (40.9%)

**Abbreviations**—1L: first line; A+B: atezolizumab plus bevacizumab; ECOG PS: Eastern Cooperative Oncology Group Performance Status; SD: standard deviation; uHCC: unresectable hepatocellular carcinoma. **Notes:** ^1^ The percentages may not add up to 100% because a small proportion of patients with unknown information has not been reported in the table. ^2^ The baseline period was defined as the period of up to 1 year prior to the initiation of A+B.

**Table 3 cancers-17-01814-t003:** Disease characteristics in patients with uHCC receiving A+B as 1L treatment.

	Overall Cohort
	*n* = 300
**Primary HCC etiology, *n* (%)**	
Viral etiology ^1^	132 (44.0%)
Hepatitis C virus	114 (86.4%)
Hepatitis B virus	17 (13.0%)
Non-viral etiology ^1^	108 (36.0%)
Non-alcoholic steatohepatitis	34 (31.5%)
Alcohol-related fatty liver disease	34 (31.5%)
Non-alcoholic fatty liver disease	9 (8.3%)
Diabetes mellitus	6 (5.6%)
Metabolic syndrome	2 (1.9%)
Other etiology ^2^	35 (11.7%)
Unknown etiology	25 (8.3%)
**CP class within 90 days of initiation of 1L A+B treatment, *n* (%)**	
A (CP score = 5–6 points)	219 (73.0%)
B (CP score = 7–9 points)	79 (26.3%)
C (CP score = 10–15 points)	2 (0.7%)
**BCLC stage ^3,4^ within 90 days of initiation of 1L A+B treatment, *n* (%)**	
A: Early	8 (2.7%)
B: Intermediate	41 (13.7%)
C: Advanced	247 (82.3%)
D: End-stage	3 (1.0%)
**HCC tumor invasion ^5^ within 90 days of initiation of 1L A+B treatment, *n* (%)**	
Extrahepatic spread	120 (40.0%)
Vp4 portal vein invasion	76 (25.3%)
Tumor invasion > 50% of liver	57 (19.0%)
Bile duct invasion	13 (4.3%)
None of the above	108 (36.0%)
**ALBI grade ^3,6^ within 90 days of initiation of 1L A+B treatment, *n* (%)**	
1	98 (32.7%)
2	173 (57.7%)
2A	86 (49.7%)
2B	87 (50.3%)
3	27 (9.0%)

**Abbreviations**—1L: first line; A+B: atezolizumab plus bevacizumab; ALBI: albumin–bilirubin; BCLC: Barcelona Clinic Liver Cancer; CP: Child–Pugh; HCC: hepatocellular carcinoma; SD: standard deviation; uHCC: unresectable hepatocellular carcinoma. **Notes:** ^1^ Some patients’ primary HCC etiology could only be established as viral/non-viral. As a result, the subcategories may not sum to 100%. ^2^ Other primary HCC etiologies included alcohol abuse, autoimmune disease, autoimmune hepatitis, cirrhosis, cryptogenic non-viral etiology, hemochromatosis, hyperlipidemia, or hypertension. ^3^ The percentages may not add up to 100% because a small proportion of patients with unknown information has not been reported in the table. ^4^ The BCLC scores collected from the study centers were adjudicated algorithmically based on the BCLC Staging System to ensure alignment with the corresponding patient’s CP class/score. Accordingly, 89 patients were recategorized as BCLC stage C and 5 patients as BCLC stage D, respectively. ^5^ Multiple responses were allowed for this question, and the percentages may add up to more than 100%. ^6^ The ALBI grade was determined as follows: grade 1: ≤−2.60; grade 2: >−2.60 and ≤−1.39; grade 2a: >−2.60 and ≤−2.118; grade 2b: >−2.118 and ≤−1.39; and grade 3: >−1.39.

**Table 4 cancers-17-01814-t004:** Treatments received prior to and following A+B as 1L treatment among patients with uHCC.

	Overall Cohort
	*n* = 300
**Locoregional HCC treatment prior to 1L A+B treatment ^1^, *n* (%)**	
No prior locoregional treatment received	144 (48.0%)
Embolization (transarterial embolization or transarterial chemoembolization)	88 (29.3%)
Transarterial radioembolization	49 (16.3%)
Ablation (includes radiofrequency ablation or microwave ablation)	43 (14.3%)
Stereotactic body radiotherapy	26 (8.7%)
Selective internal radiation therapy	7 (2.3%)
External beam radiotherapy	1 (0.3%)
Other locoregional treatment received ^2^	10 (3.3%)
**Systemic HCC therapy following 1L A+B treatment ^1^, *n* (%)**	
No systemic treatment received	208 (69.3%)
Lenvatinib	56 (18.7%)
Cabozantinib	15 (5.0%)
Sorafenib	13 (4.3%)
Pembrolizumab	11 (3.7%)
Nivolumab	7 (2.3%)
Regorafenib	4 (1.3%)
Ipilimumab	5 (1.7%)
Other systemic therapy received ^3^	26 (8.7%)

**Abbreviations**—1L: first line; A+B: atezolizumab plus bevacizumab; HCC: hepatocellular carcinoma; uHCC: unresectable hepatocellular carcinoma. **Notes:** ^1^ Multiple responses were allowed for this question, and percentages may add up to more than 100%. ^2^ Other locoregional treatments used prior to 1L A+B included yttrium-90 embolization, cryoablation, proton radiation therapy, proton therapy, robotic assisted left lateral sectionectomy, and intraoperative ultrasounds of liver. ^3^ Other systemic therapies used after 1L A+B included cabozantinib, denosumab, FOLFOX, palliative radiation therapy, tremelimumab/durvalumab, capecitabine, HFB30100, cyclophosphamide/fludarabine, durvalumab, futibatinib, futibatinib and pembrolizumab, futibatinib/durvalumab, gemcitabine, gemcitabine/bevacizumab, trastuzuamab, and zoledronic acid.

**Table 5 cancers-17-01814-t005:** Clinician-assessed best treatment response among patients with uHCC receiving A+B as 1L treatment.

	Overall Cohort
	*n* = 300
**Clinician-assessed best treatment response ^1^, *n* (%)**	
rwCR	22 (7.3%)
rwPR	70 (23.3%)
rwSD	118 (39.3%)
rwPD	59 (19.7%)
Unable to assess	31 (10.3%)
**Duration of best treatment response (mean ± SD; months)**	10.3 ± 8.3
**Radiological assessment types used to assess best treatment response ^2^**	
Computed tomography	177 (59.0%)
Magnetic resonance imaging	157 (52.3%)
18F fludeoxyglucose positron emission tomography	6 (2.0%)
Magnetic resonance spectroscopy	2 (0.7%)
Other	1 (0.3%)
Unknown	29 (9.7%)

**Abbreviations**—1L: first line; A+B: atezolizumab plus bevacizumab; RECIST: Response Evaluation Criteria in Solid Tumors; rwCR: real-world complete response; rwPR: real-world partial response; rwPD: real-world progressive disease; rwSD: real-world stable disease SD: standard deviation; uHCC: unresectable hepatocellular carcinoma. **Notes:** ^1^ A patient’s best treatment response was defined as their best clinician-assessed disease status, identified from up to eight radiological assessments following treatment initiation with A+B; the use of RECIST methodology was not required. ^2^ Multiple responses were allowed for this question, and the percentages may add up to more than 100%.

## Data Availability

Due to the nature of this research, the participants of this study did not agree for their data to be shared publicly, so supporting data are not available. Therefore, restrictions apply to the availability of these data, which are not publicly available.

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
