# Peer review of "Real-World Experiences Using Atezolizumab + Bevacizumab for the Treatment of Unresectable Hepatocellular Carcinoma: A Multicenter Study†"

_cancers, 2025, doi:10.3390/cancers17111814_

Round 1

Reviewer 1 Report

Comments and Suggestions for Authors

This retrospective, multicenter chart review investigates the real-world effectiveness of first-line atezolizumab plus bevacizumab (A+B) in treating unresectable hepatocellular carcinoma (uHCC) across five U.S. academic centers. The study includes a diverse cohort of 300 patients, including those underrepresented in clinical trials, such as individuals with Child-Pugh class B liver function and racial/ethnic minorities. Clinical outcomes—including real-world progression-free survival (6.8 months) and overall survival (14.4 months)—were consistent with prior trial data, especially within a “trial-like” subgroup. Hospitalization rates were higher in patients with impaired liver function, though toxicity-related adverse events were similar across groups. The study is strengthened by its focus on real-world clinical diversity and healthcare resource utilization, contributing valuable evidence to support A+B's use outside controlled trial settings.

Some suggestions:

  • better betail how patients were screened and selected to ensure reproducibility and transparency.

  • Consider incorporating RECIST or mRECIST criteria for treatment response to improve comparability with existing literature.

  • Provide more detail on how missing data were handled and clarify censoring rules for survival analyses.

  • Expand the limitations section, particularly regarding retrospective data quality and inter-center variability. Furthermore, please tone down some statements (e.g., abstract, line 47)
  • Consider conducting subgroup or sensitivity analyses to test the robustness of findings

Author Response

Thank you for the comments.  Please see attached response.

Reviewer 2 Report

Comments and Suggestions for Authors

Authors showed the results of real-world data of 1L A+B therapy in US population. This paper was well-addressed and well-written. It can provide useful information for readers. But some issues remain to be addressed.

  1. In clinical setting, combined loco-regional therapies such as TACE or radiation might be performed. If so, the data should be shown.
  2. Regarding with treatment-related AEs, 8.8% TRAEs were reported in this study. The incidence might be lower. The incidence of AEs requiring steroid should be clarified. Those of bevacizumab-induced AEs should be also clarified. In real-world setting, patients with comorbidities were included. Management of TRAEs is important for such population.

Author Response

(The authors gave the same response as above.)

Reviewer 3 Report

Comments and Suggestions for Authors

Note is made that this is a retrospective analysis of pooled data sourced from 5 different centers. Hence it does raise some additional questions. These are as follows-

1) The eligibility criteria for the patients to undergo treatment with A & B that are outlined in the Methods, seem quite broad. Is it possible that the acceptance criteria did in fact differ between the units based on what criteria were used to rule out patients from receiving the therapy?

2) Also, the baseline period mentioned in the Methods section does seem quite long, is it possible that this also led to some confounding of the results, noting the biological nature of this malignancy in clinical practice (and the large numbers of patients in the study cohort with advanced disease)?

3) Is anything known about whether the patients with hepatitis C had been treated with antivirals or not historically prior to this therapy being considered (i.e. was the Hepatitis C active or inactive and was this a factor in their management)?

Author Response

(The authors gave the same response as above.)

Reviewer 4 Report

Comments and Suggestions for Authors

Review on the manuscript titled “Real-world experience with atezolizumab + bevacizumab for the treatment of unresectable hepatocellular carcinoma: A multicenter study” by Abdelrahim et al., 2025.

                The authors compile their research of (unresectable) hepatocellular carcinoma treatment (uHCC) with atezolizumab and bevacizumab across 5 major US clinics, namely: Mayo Clinic, Houston Methodist, Moffitt Cancer Center, Mays Cancer Center (University of Texas Health, San Antonio), and University of Arizona, encountering 300 patients overall.

                The introduction of the study represents overall review on the subject. The authors noted that “in the past two decades, evolving from tyrosine kinase inhibitors (TKIs; e.g., sorafenib, lenvatinib) to immunotherapy-based treatments as the current standard of care in the first-line (1L) setting”. May 2020 FDA approval of “atezolizumab plus bevacizumab (A+B) for the treatment of adult patients with uHCC”, underscores the relevance of their study in presenting the real-world current data (RWD) on the subject.

                After outlining the basic status on the subject in introduction, the authors present the M&M section:

  • Study design and data source subchapter: the authors list their patient resource institute, and mention the routine treatment of patients’ data.
  • Study population subchapter criteria: 5 accepting criteria were outlined on the patients:

1) confirmed diagnosis of uHCC; 2) had no current or prior diagnosis of other primary malignancies (except listed ones) within 5 years of A+B initiation; 3) had initiated treatment with A+B for uHCC since January 1, 2019, outside of a clinical trial setting; 4) were 18 years

of age or older at the time of A+B initiation.

  • Study measures and outcomes subchapter: the authors ascertained states and conditions in their assessment of measures and outcomes.
  • Subgroups: The authors outline 2 subgroups of patients in their study cohort based on relevant criteria mentioned within.
  • Statistical analysis subchapter outlines stat. methods and the parameters they assessed in the study.

Result section.

1). Patient characteristics. The chapter outlines:

Table 1, “Demographic characteristics of patients with uHCC receiving A+B as 1L treatment“ featuring study cohort, including:

  1. a) Medical centers patients percentage breakdown (5 centers);
  2. b) Patient (average) age at initiation of 1L A+B1 (mean ±SD; years); sex, race (4 races), major ethnicity (Hispanic/not Hispanic); Education, Employment status and Insurance type.

Table 2.” Clinical characteristics of patients with uHCC receiving A+B as 1L treatment” featuring multiple clinical traits patients distribution.

Table 3. “Disease characteristics in patients with uHCC receiving A+B as 1L treatment”, including 1) ‘Primary HCC etiology’; ‘CP class within 90 days of initiation of 1L A+B treatment’,

2) ‘CP class, within 90 days of initiation of 1L A+B treatment’,

3) ‘BCLC stage3,4, within 90 days of initiation of 1L A+B treatment’,

4) ‘HCC tumor invasion5, within 90 days of initiation of 1L A+B treatment’,

5) ‘ALBI grade3,6, within 90 days of initiation of 1L A+B treatment’  patients distribution.

Additional traits are presented in Supplementary Tables S1 and S2.

2)  Treatment discontinuation chapter.

                The chapter features Fig. 1 plot of patients’ treatment discontinuation within 24 months period.

3) ”Receipt of EGD and HCC treatments prior to and after A+B” results reported in Table S3, while scheme of EGD assessments patients distribution is presented in Fig. 3.

Table 4. Treatments received prior to and following A+B as 1L treatment, among patients with uHCC outlines statistics on HCC.

4) Section “Real-world effectiveness outcomes” features several subchapters:

  1. a) Clinician-assessed treatment response (Table 5).
  2. b) Fig. 2 “Median rwPFS (95% CI) = 6.8 (5.8, 8.4) months” plot outlining results.

5) “Overall survival (OS)” chapter presents the OS plot in Fig. 3.

6) “HCC-related hospitalizations and ER visits” presents the table on ER visits (Table S4) and scheme 4 outlining “HCC-related healthcare resource utilization in patients with uHCC receiving A+B as 1L treatment, during the first year of treatment with A+B”.

After the explicit and comprehensive discussion on the results, the authors outlined their major conclusion as:

“This study demonstrates that A+B is an effective treatment in routine clinical practice.” The authors also note that a significant part of the patients were excluded due to the accompanying diagnosis exempting them from the current study.

                Overall, the manuscript addresses quite an actual issue on the expanded sample maintained by outstanding medical centers, and would be of significant interest to the clinicians in the field. The structure of the manuscript is a bit complicated and may render some work (not compulsory). Commercial funding mentioned in acknowledgements. Some notes are presented below.

  • Supplementary Tables: didn’t find it supplied.
  • It’s not clear what’s going on with Scheme tables: Scheme 1 and 2 tables are located after bibliography without any referencing in the text. Are they going into supplement? Please, arrange/reference them accordingly. It’s not exactly clear how scheme term differs from table in your instance.

Author Response

(The authors gave the same response as above.)

Round 2

Reviewer 3 Report

Comments and Suggestions for Authors

Note has been made of the responses provided by the authors to the reviewers questions along with the alterations that have been made to the manuscript